# Perceptions of and Attitudes toward COVID-19 Vaccination among Urban Slum Dwellers in Dhaka, Bangladesh

**Wafa Alam** [1,*] , **Nadia Farnaz** [1] , **Farzana Manzoor** [1] , **Sally Theobald** [2] **and Sabina Faiz Rashid** [1]

1   BRAC James P Grant School of Public Health, BRAC University, 6th Floor, Medona Tower,
    28 Mohakhali Commercial Area, Bir Uttom A K Khandakar Road, Dhaka 1213, Bangladesh
2   Department of International Public Health, Liverpool School of Tropical Medicine, Pembroke Place,
    Liverpool L3 5QA, UK
*   Correspondence: wafa.alam@bracu.ac.bd or wafa.alam94@gmail.com

**Abstract:** Vaccine hesitancy or low uptake was identified as a major threat to global health by the World Health Organization (WHO) in 2019. Vaccine hesitancy is context-specific and varies across time, place, and socioeconomic groups. In this study, we aimed to understand the perceptions of and attitudes toward COVID-19 vaccination through time among urban slum dwellers in Dhaka, Bangladesh. In-depth telephone interviews were conducted between October 2020 and January 2021 with 36 adults (25 females and 11 males) living in three urban slums of Dhaka City, Bangladesh. Follow-up interviews were undertaken in April and August 2021 to capture any shift in the participants' perceptions. Our findings show that for many there was an initial fear and confusion regarding the COVID-19 vaccine among people living in urban informal settlements; this confusion was soon reduced by the awareness efforts of government and non-government organizations. Women and young people were more interested in being vaccinated as they had had more exposure to the awareness sessions conducted by non-governmental organizations (NGOs) and on social media. However, people living in the slums still faced systemic barriers, such as complicated online vaccine registration and long queues, which led to low uptake of the vaccine despite their increased willingness to be vaccinated. This study highlights the importance of using sources such as NGO workers and television news to debunk myths, disseminate COVID-19 vaccine information, and support adherence to vaccination among urban slum dwellers. Our study underscores the importance of addressing systemic barriers blocking access and understanding community perceptions in order to develop effective communication strategies for vulnerable groups that will then improve the COVID-19 vaccine uptake.

**Keywords:** COVID; COVID-19 vaccines; urban slums; informal settlements; vaccine hesitancy; contextual factors



## 1. Introduction

The COVID-19 pandemic has infected over 647 million people and has caused 6.6 million deaths around the globe, as of 8 December 2022 (Johns Hopkins University n.d.). COVID-19 has had devastating effects on families, communities, and the economy globally, with the resulting burden falling disproportionately on vulnerable groups, including healthcare workers, elderly people, low-income communities, and other marginalized groups (Goldstein 2020; PAHO 2020). Public health measures, such as lockdowns and social distancing measures, have impacted people's social lives and economic activity, further triggering a mental health crisis (Al Dhaheri et al. 2021; Bodrud-Doza et al. 2020; Das et al. 2021; Xiong et al. 2020). Scientific communities around the world are working to develop vaccines against the virus. To date, 92 vaccines have advanced to phase 3 of the vaccine trials (Basta and Moodie 2020a), while another 50 vaccines have been approved across multiple countries—for example, the Pfizer/BioNTech vaccine and the Oxford-AstraZeneca vaccine have been approved in 149 countries around the world (Basta and Moodie 2020b).

Vaccine hesitancy is defined by the WHO as a "delay in acceptance or reluctance or refusal of vaccines, despite the availability of vaccination services" (MacDonald 2015) and has been identified as one of the ten major threats to global health in 2019 (World Health Organization (WHO) 2019). Some commonly identified factors shaping vaccine acceptance include opinions regarding the efficacy of the vaccine, a lack of trust in the health system, a lack of knowledge in communities, misinformation and misconception about the need for vaccination, etc. (Dubé et al. 2013; Oduwole et al. 2021; Setbon and Raude 2010). The same challenges apply to the COVID-19 vaccine, as shown by studies conducted in China and Saudi Arabia, which reported perceived risk and trust in the health system as factors that determine vaccine acceptance (Narapureddy et al. 2021; Solís Arce et al. 2021; Wang et al. 2020).

In Bangladesh, around 47 percent of the urban population lives in informal settlements or slums (The World Bank 2018). These contexts are densely populated, with the majority of households living in one room and using a shared kitchen, toilet, and water source, thus lacking adequate water, sanitation, healthcare, schooling, etc. (Bangladesh Bureau of Statistics 2015; Haque et al. 2020). People in slums live and work in poor conditions, making them more prone to diseases and health burdens. They are often considered invisible, with no rights and little in the way of a voice to influence policies that would work to their benefit. It is important to ensure that people living in urban slums are prioritized when the COVID-19 vaccine becomes available (Lines et al. 2022). The risk factor for COVID-19 transmission is high in these areas, due to overcrowding and the inability to practice preventive measures such as social distancing (Friesen and Pelz 2020; Rashid et al. 2020).

The urban areas of Bangladesh, particularly the slums, suffer from the lack of a well-defined health service provision system. The Ministry of Health's primary health care services do not cover the urban poor, who fall under the responsibility of the local government. However, the local government lacks the necessary resources and infrastructure to provide adequate healthcare services in urban areas (Albis et al. 2019). Primary healthcare services in these urban areas are provided through the Urban Primary Healthcare Services Delivery Project, which contracts non-governmental organizations (NGOs), and via government dispensaries. These facilities face numerous challenges, including insufficient accountability, weak monitoring systems, staff shortages, and limited services (Albis et al. 2019). The absence of an umbrella model has led to a lack of coordination and standardization among these healthcare facilities, further exacerbating the challenges faced by the urban poor when accessing healthcare services, leading to an overall reliance on drugstore sellers for medicines and to access support and advice (Afsana and Wahid 2013).

As in most countries, Bangladesh also rolled out vaccination against COVID-19, along with an initial 5 million Covishiled vaccine doses from India, commencing 7 February 2021 (Hassan 2021). Initially, the Government of Bangladesh published a priority list for vaccination, which included frontline health workers and people aged 40 years and over for the first round. Soon, the vaccination program was opened up to other population groups and is now open to all adults; it is also being administered in schools for school-going children. Those willing to be vaccinated register through a mobile and web-based application called "Surokkha", which means "protection" in Bengali. This web portal and mobile application was developed by the ICT division of the Government of Bangladesh to facilitate the registration of COVID-19 vaccines for its people. In Bangladesh, COVID-19 vaccines were administered at tertiary healthcare centers in Dhaka and also through district hospitals and Upazila Health Complexes[1] in other parts of the country.

The government of Bangladesh also rolled out week-long mass vaccination campaigns across many unions, municipalities, and city corporations from 8 August 2021 (Islam and Shishir 2021). A second phase of mass COVID-19 vaccinations was rolled out in late March 2022, with the target of vaccinating 90% of the country's population (UNB 2021). These allowed people who could not register for the vaccinations to use a web-based application to access the vaccines. Registration was performed on the spot, using national identity cards or birth certificates. The government had also brought slum dwellers under the

umbrella of a vaccination campaign, with the aim of immunizing citizens who were not vaccinated during the mass vaccine campaigns. The urban slum campaign started in Korail, the largest slum in Dhaka, and was then expanded to other slums (Kamruzzaman 2021). As of 24 March 2023, over 357 million vaccines have been administered in the country (WHO n.d.). Around 88% of the country's population has received one dose, while around 80% has received 2 doses of the vaccine to date (DGHS 2023)

Quantitative studies in Bangladesh have shown a low vaccine acceptance rate among urban slum dwellers (58.1%) compared to other geographic locations, such as rural and peri-urban areas (Mamun and Fatima 2021), and that slum dwellers are not interested in taking the vaccine (Abedin et al. 2021). Exploring these issues is of global importance as delays in vaccination programs can lead to the spread of new variants (Sachs et al. 2021). Vaccine acceptance is context-specific and varies across places, cultures, and different socioeconomic groups (Larson et al. 2018; Xiao and Wong 2020). Therefore, having a nuanced and context-embedded understanding of the different attitudes and experiences of urban slum dwellers with respect to the COVID-19 vaccine is critical.

Therefore, we aimed to understand the perceptions, experiences, and challenges faced in COVID-19 vaccination among urban slum dwellers in three slums in Dhaka city, where the research team has been working since 2019. One needs to apply a broader framework and understand the social and economic fabric of people's lives, along with the multiple factors at play, when unpacking exactly what is meant by hesitancy in these diverse contexts of slums and deprived areas.

## 2. Methods

### 2.1. Study Design

A qualitative study using in-depth telephone interviews was conducted to explore and gain a more profound understanding of the perceptions and experiences related to COVID-19 vaccines among slum dwellers living in Dhaka during the period from October 2020 to August 2021 (from first awareness of the vaccines to the vaccination program rollout). This temporal data collection and the subsequent analysis were important as the evolution of the COVID-19 vaccine program was dynamic and fast-changing. Data for this paper were drawn from a COVID-19 case study conducted as a part of a larger research project named ARISE[2], which focuses on the health and wellbeing of urban slum dwellers.

### 2.2. Study Setting and Sampling Procedure

This study was conducted in three urban slums in Dhaka (Table 1). These three slums were purposefully selected to capture diversity in terms of geography and service provision, based on our previous research and relationships.

**Table 1.** Characteristics of the three study sites.

| Characteristics | Slum A | Slum B | Slum C |
|---|---|---|---|
| Year of establishment | 1988 | 1991 | 1996 |
| Location | Dhaka North City Corporation | Dhaka South City Corporation | Dhaka South City Corporation |
| Built on | Government land | Government land | Private land |
| Population | 15,750 people | 20,000 people | 6800 people |

**Table 1.** *Cont.*

| Characteristics | Slum A | Slum B | Slum C |
|---|---|---|---|
| NGO services that were stopped during the lockdown | BRAC UDP: Livelihood support, WASH program Caritas Bangladesh: Education program Dhaka Ahsania Mission: Education program ActionAid: Gender-based violence and childhood development | BRAC UDP: Livelihood support, WASH program Surovi: Education program PLAN International: Education program | No NGOs were working in this community before the COVID-19 pandemic |
| NGO services provided during the lockdown | BRAC UDP: Increasing awareness, relief distribution, cash support, hygiene product distribution DSK: Food and cash support Dhaka Ahsania Mission: Food support | BRAC UDP: Increasing awareness, relief distribution, cash support, hygiene product distribution Plan International: Food support, hygiene product distribution | BRAC UDP: Increasing awareness, relief distribution, cash support, hygiene product distribution |
| NGO services being provided now | BRAC UDP: Livelihood support, WASH program, food support in collaboration with the World Food Program Caritas Bangladesh: Education program and daycare ActionAid: Gender-based violence and childhood development DSK: WASH program | BRAC UDP: Livelihood support, WASH program Surovi: Education program PLAN International: Education program, vaccine registration support SAJIDA Foundation: Vaccine registration support | BRAC UDP: Increasing awareness, relief distribution, cash support, hygiene product distribution, vaccine registration support, health camps |

Snowball and opportunistic sampling techniques were used for the recruitment of study participants. Program personnel from organizations operating in the study sites were contacted to collect the phone numbers of contact persons (usually, community health workers (CHWs)) within the respective study sites. These CHWs, who mainly live and work in these slums, were our first point of contact for recruiting participants. Additional informal discussions were conducted by phone with key influential people from within urban slums, such as community leaders and community organization members, to explore the residents' perceptions of marginality. A list of respondents under the specified categories—extremely poor households, women-headed households, people with disabilities, new migrants, and comparatively well-off households (Table 2)—was prepared by members of the research team for each study site, and phone numbers were accessed. It was estimated that around 30 interviews would be needed to achieve rich and saturated data. Factors such as the length of stay in the slums, gender, and socioeconomic condition were considered in the sampling approach to obtain diversity in participant perceptions regarding COVID-19 vaccine uptake.

**Table 2.** Descriptions of the respondent categories.

| Most Marginalized Groups | Description |
|---|---|
| Pregnant and Lactating Mothers | Women (aged 18 and above) who are currently pregnant or lactating, with a child of less than 6 months old. This group is identified as a highly marginalized group as their earning opportunities may be compromised by pregnancy/and or child-caring responsibilities. |
| Person with Disabilities | A person who has either a physical or mental disability or both. They have limited mobility and are often dependent on other family members/neighbors/relatives for their livelihood, performing daily household and personal activities, and accessing services. |

**Table 2.** *Cont.*

| Most Marginalized Groups | Description |
| --- | --- |
| Single-female-headed Household | Households headed by a female member who is the only earning member and the main decision-maker of the family, for example, households headed by women who are widowed/divorced/abandoned by their husbands. These women are mostly involved in the informal economy sector and are considered marginalized as there is no male family member earning money to support them, which may put these families in a socially vulnerable position. Female members may be more at risk of sexual and gender-based violence. |
| Informal Worker (Male) | The community considers a family marginalized if the male member of the family is involved in the informal economy sector, such as daily wage earners who do not have a legally binding appointed job in a legal organization (government/private/NGO). A lack of a fixed earning source may make these individuals and families prone to poverty. |
| Elderly People | Elderly men and women aged 60 and above are also considered a marginalized group because they often have limited mobility and are also financially dependent on others. |
| Comparatively better-off individuals | Individuals who are not dependent (financially or otherwise) on others and have a relatively stable income. |

### 2.3. Data Collection

The pre-tested interview guidelines contained questions related to perceptions and practices throughout the COVID-19 pandemic and its impact on the participant's personal, economic, and social life. Individual in-depth repeated telephone interviews with 30 individuals (8 males and 22 females) were conducted from October 2020 to January 2021 by the members of the research team, who had prior experience of conducting qualitative interviews with slum dwellers.

Each interview was repeated with the individuals and conducted over 3–4 sessions, with each session ranging between 45 and 60 min. The dates and times for the interviews were organized at the convenience of the respondents. This meant having to reschedule two to three times to accommodate other commitments, as well as conducting interviews during the evenings or at weekends, particularly in the case of male respondents who worked outside their homes during the day.

Following the launch of the COVID-19 vaccination campaign in Bangladesh, follow-up telephone interviews were conducted with an additional seventeen respondents from April to May 2021 to understand any changes in the information recorded or in perceptions and practices about COVID-19 vaccines. Another nine telephone interviews were conducted in August 2021 with our study participants and community researchers/co-researchers after the mass vaccination campaign, which took place throughout the country.

All interviews were audio-recorded and informed verbal consent was sought from all participants prior to the telephone interviews. The recordings were transcribed and translated from Bengali to English by a team of trained transcribers within a week of data collection and were then reviewed by the respective interviewers for accuracy. All data were securely stored on a Google drive that could only be accessed by the research team.

### 2.4. Data Analysis

Manual inductive coding and thematic analysis were conducted by a group of trained researchers. Sections of text were organized into open codes that emerged inductively from the data during the preliminary analysis, using phenomenological and grounded theory. The codes were then grouped under different themes and sub-themes, using discursive techniques to identify the question patterns, assess context, and achieve conceptual clarity (Nowell et al. 2017). Discussions among the research team helped reach a consensus on the key themes, based on preliminary analysis and the coding and clustering of the data.

Transcripts were coded independently by each researcher, followed by peer debriefing to ensure reliability and credibility.

### 2.5. Ethical Approval

Ethics approval for this research was received from the Institutional Review Board of the BRAC James P. Grant School of Public Health, BRAC University, Dhaka (IRB Reference No. 2019-034-IR). The study objectives and the voluntary nature of the study were clearly explained to the participants and informed verbal consent was taken at the beginning of each telephone interview session. Confidentiality was maintained by using unique identification numbers. Each respondent was given monetary compensation of BDT 300 (USD 2.96) at the end of their final interview session, transferred via mobile banking applications, in recognition of giving their time up to be interviewed.

### 3. Results

The participants' characteristics, such as their age, marital status, occupation, family size, and income are shown in Table 3. Interviews were conducted with both male and female respondents aged 18 and above. However, we deliberately identified more women than men as other studies have shown that women and girls are disproportionately affected in times of crisis, such as during a pandemic (Arendt 2020), and their voices tend to be unheard.

**Table 3.** Characteristics of the study participants across the different phases of the research.

| Characteristics | | Participants (October 2020–January 2021) (*n* = 30) | Participants (April–May 2021) (*n* = 17) | Participants (August 2021) (*n* = 09) |
|---|---|---|---|---|
| Gender | Male | 8 | 7 | 4 |
| | Female | 22 | 10 | 5 |
| Marital status | Married | 21 | 15 | 5 |
| | Widow | 3 | 1 | 1 |
| | Divorced/separated | 4 | 0 | 0 |
| | Unmarried | 2 | 1 | 3 |
| Religion | Muslim | 30 | 15 | 9 |
| Age Range | | 18–80 years | 20–51 years | 18–51 years |
| Average Family Size | | 5 | 5 | 5 |
| Occupations of the respondents | Domestic help | 13 | 5 | 1 |
| | Housewife | 4 | 2 | 0 |
| | Small business owner | 3 | 1 | 1 |
| | NGO worker | 2 | 2 | 0 |
| | Cook | 1 | 1 | 1 |
| | Factory/RMG worker | 2 | 0 | 0 |
| | Street hawker | 1 | 2 | 0 |
| | Hotel worker | 0 | 1 | 0 |
| | Day laborer | 1 | 2 | 1 |
| | Beggar | 1 | 0 | 0 |
| | Retired | 1 | 0 | 0 |
| | Youth volunteer | 1 | 1 | 5 |
| Family's monthly income | <5000 taka (<USD 58) | 3 | 4 | 2 |
| | 6000–10,000 taka (USD 70–119) | 15 | 5 | 3 |
| | 11,000–20,000 taka (USD 120–235) | 15 | 5 | 3 |
| | >20,000 taka (>USD 235) | 3 | 3 | 1 |

The findings are grouped according to the following themes that emerged from the analysis, in the order of the chronology of events that took place from late 2020 to the middle of the year 2021: (1) initial (pre-roll out) perceptions about COVID-19 vaccines (2) the COVID-19 vaccine rollout, continued fears, anxieties about access and about new and

emerging rumors, (3) community mobilization and the increase in vaccine uptake, and (4) the remaining systemic and other barriers to vaccine uptake.

### 3.1. Initial (Pre-Rollout) Perceptions about COVID-19 Vaccines in Communities

During the period covering the first set of interviews (October 2020 to early January 2021), limited information was available on COVID-19 vaccination. Most people (25) in all three study sites had heard about the COVID-19 vaccine from the television news. As a 35-year-old widow from Slum A, who works as a part-time housemaid in two households, explained:

> "I didn't hear much about the vaccine. My madame told me about the vaccine once. She saw on the TV that there will be an injection coming to prevent coronavirus. She said she would also make me take the vaccine to be protected. However, when her father-in-law got infected with the virus, I left the job".

Those who had heard about the COVID-19 vaccine knew that a vaccine for coronavirus has been invented and was beginning to be rolled out in other countries. There were various opinions on vaccine rollout in Bangladesh, including the theories that the government of Bangladesh was trying to source the vaccine, that Bangladesh would receive the vaccine from India, that the vaccine would come to Bangladesh by March 2021, that Bangladesh and India would be the last countries in the world to receive the vaccines, and so on. The participants' sources of information included the news programs on television and social media platforms, along with word of mouth from tea-stall conversations.

Many of our respondents (24) were willing to take the vaccine if it was accessible to them and also, preferably, free of charge. At this point in time, it was unclear to them if the vaccine would be available for free. The respondents understood the importance of the vaccine in helping to protect themselves from diseases. This trust in the vaccines appears to be linked to the Bangladeshi Expanded Program on Immunization (EPI), a successful public health intervention that has contributed to reducing mortality and morbidity from many preventable diseases such as polio, measles, etc. (Sarkar et al. 2015). This trust was commonly found among men and women from all three sites and illustrative quotes include the following.

A 38-year-old female part-time housemaid living in Slum A, with two young children, said: "Well I will take the COVID vaccine! In our country there was polio before, there was diphtheria, there were different types of diseases, but now these diseases are gone. The government has provided vaccines for these diseases. When we women were pregnant, we were given the TT vaccine from the hospital. That is why I will take the vaccine so that I won't be infected, my children won't be infected. Only if I can afford it."

Another 80-year-old retired male respondent from Slum C shared similar views: "Yes, of course I will take the vaccine. Some time ago, there were no vaccines for other diseases also, but now there is. So, people die less than before. So, we should take the corona vaccine also."

Prior to the COVID-19 vaccine rollout in Bangladesh, there were different opinions and varying degrees of confusion among the general population of the country regarding which population groups were prioritized for the vaccine, as well as concerns about its cost, efficacy, and safety. Some key emerging ideas were as follows: only the rich would be vaccinated; the government staff would be prioritized; others, such as those living in slums, would be neglected.

### 3.2. COVID-19 Vaccine Rollout, Continued Fears, Anxieties about Access, and New and Emerging Rumors

When followed up from April to May 2021, people at all three sites had heard about the introduction of the COVID-19 vaccine in Bangladesh, and many knew that the vaccines were coming from foreign countries such as India and China. Other information about the COVID-19 vaccine and the registration process was not very widespread and television news programs were still the major source of information at this point. Most people knew

very little about the vaccine registration process and had heard negative reports of COVID-19 vaccination side effects. They had heard on the television or from people at the tea stalls and in the marketplace about the need to register for the vaccine using their phones and the internet, but confusion was still present. A 20-year-old female from Slum A, working part-time as a household help, said: "Everyone can take the vaccine. People like us can also take the vaccine. But I don't know where this vaccine has come from. In our area, vaccines have not come yet . . . but I heard [that] in other areas [districts], people have received it."

Few people in the community were being vaccinated. Fear, confusion, and misconceptions were fairly widespread; while some were willing to take the vaccine, others were scared and were worried about the safety and efficacy of the vaccine as they had heard many different rumors via multiple personal and social conversations about the vaccine.

### 3.2.1. Rumors about Fake COVID-19 Vaccines and Fears of Side Effects and of Death

A few participants expressed the fear of being fooled into taking a fake vaccine and rumors were circulating in communities regarding the vaccine that came from India (named COVISHIELD), the first COVID-19 vaccine that was administered in Bangladesh. People had heard all kinds of comments from others in tea stalls or the marketplace, such as rumors that people were being injected with saline water instead of the vaccine, or that the vaccine had some ingredients deliberately mixed into it that would cause death. These views were informed by the news or by hoax discussions circulating in social media about the vaccine that was manufactured by India.

One 20-year-old part-time housemaid from Slum A said:

"People think something will be mixed with vaccines. Something else will be introduced along with vaccines and people will get diseases if they take the vaccine. That's why some people don't want to take the vaccine".

Another 39-year-old NGO field worker from Slum B explained:

"You will hear many people say many things. Some are saying that there is no value in these vaccines. It is all saline water. People will say what they have to, what can you do? People that don't understand the situation or people that are ignorant [of] this, say it. One person was telling me that the Government went to India and got saline water and are giving saline water to all of you. They are saying this from a lack of trust but it's not all of them, it's only some of them".

One major fear among these people regarded reports of deaths caused by the COVID-19 vaccine, as discussed on Facebook, YouTube, and some TV channels. While the people reported that they did not believe these rumors, the rumors continued to circulate, including that "people die within eight months of taking the vaccine", "whoever is taking the vaccine is dying", etc. These rumors created fear among some slum community members, bringing anxiety that led them to be hesitant to register for vaccination.

### 3.2.2. Misconceptions about the People for Whom the COVID-19 Vaccine Is Intended

There were some misconceptions in the community about the COVID-19 vaccine. These were particularly prevalent in Slum C, where fewer NGO services were operating. Two women from Slum C mentioned the rumor that the COVID-19 vaccine was particularly intended for children; for example, a 45-year-old female homemaker from Slum C said:

"I heard they give vaccines to prevent corona from the Chairman's office . . . They say this vaccine is only for children. I heard about this from the house next door. There is a mother who went and gave it [to her child], she then said to me that [her] auntie told Sharmin apa [the respondent's daughter] to give it to her kids. Then my daughter went to get it too, the day [people with corona vaccines] came, she couldn't go. So, she went the day after; later on, they didn't come, they only came that one day."

This perception that the vaccination was meant only for children could stem from Bangladesh's successful, 25-year-long, and infant-based Extended Program of Immunization. Although this was not a widespread belief, as informed by our research, it is important to ensure that this topic is properly clarified in future vaccine communications.

Another misconception that existed in the community, also specifically in Slum C, was that the COVID-19 vaccine was meant for only people who currently had COVID-19 and was not intended for everyone. A 45-year-old woman from Slum C, who was previously working in a garment factory, said:

> "We don't need it; those who have corona, they need the vaccine".

### 3.3. Community Mobilization and the Increase in Vaccine Uptake

The interviews conducted in April and May 2021 revealed that most residents from across all three study sites had not been vaccinated, even though vaccine registration and rollout had already begun, and the vaccine was being administered free of charge. There were some who did not want to accept the vaccines because they were afraid of side effects. As more people were vaccinated, this fear of side effects was slowly diminishing. Yet, for many, these fears outweighed the perceived benefits of the vaccine.

A 39-year-old NGO field worker from Slum B said:

> "My parents know about this vaccine registration, but they are very old, and they did not want to take it as they heard a lot of people got sick after taking it, they had headaches and fever so that's why they didn't want to risk it. They were scared of the side effects ... They say that they are fine now. People get sicker after taking the vaccine, so they don't want to take it."

It was during this time that community health workers from several NGOs, such as BRAC, the SAJIDA Foundation, and Plan International, played the very important role of combatting these fears and the myths surrounding the COVID-19 vaccines. They conducted community yard meetings or went from door to door, explaining to the slum dwellers that these side effects were common for any vaccines and were not a cause for concern. Community health workers also raised awareness among people about the vaccine registration system; many held programs or camps where they helped people register for the vaccine through the "Surokkha" application. These community mobilization activities were more frequent in Slum A and Slum B, compared to Slum C where just one NGO was operating in the community. People in Slum A and Slum B knew more about the vaccination program, whereas, in Slum C, no male respondents that we spoke to knew about the vaccination program. Women in Slum C learned about the vaccination drive from community and yard meetings that were arranged by BRAC, which were mainly attended by women during the day, when the men were working at the nearby mills and factories nearby.

People affiliated with NGOs, either as direct employees or as community volunteers, knew more about the registration process compared to others in the area. This finding shows that having strong social networks within the community can be useful in vaccination programs.

One of the respondents, a 39-year-old NGO field officer, said:

> "I heard the registration is done online. There is an online platform where you register for vaccines and then you get a document from them. You take the document to the vaccination center, and you get the vaccine."

The final follow-up in-depth interviews, conducted in August 2021, revealed a major shift: more people were eager to be vaccinated. This trend was seen at all three study sites. People who were reluctant to receive the vaccine earlier were now eager to be vaccinated. This change is linked to an increase in mass awareness, advertisements on television, compulsory obligations at workplaces, interpersonal communication, seeing others receiving the C-19 vaccines, and the active involvement and participation of trusted people, such as CHWs, youth volunteers, and influential actors from the study sites in the

mass vaccination programs. However, vaccine uptake was still found to be far lower in Slum C, compared to Slums A and B.

### 3.3.1. Neighbors and the Local Youth Help People in the Community Register for the COVID-19 Vaccine

Some of the respondents living in Slum A and Slum B said that people from nearby households, particularly the youths, who are more familiar with technology and have access to the internet and smartphones, helped their neighbors with vaccine registration. In Slum C, the networks within communities are not as strong as those seen in the other two slums, which may be why none of the respondents from that site reported receiving help from their neighbors regarding vaccine registration. This is because the majority of people living in Slum C worked in factories as informal laborers and were mostly living as temporary migrants, thus lacking the strong social connections that can be built over many years. A 35-year-old woman from Slum B, working as a cook, said:

> "There is a man called Rumel bhai [a pseudonym], who lives next door. He also comes to tutor my son. Noba [a pseudonym], our Community Organizer [a cadre of CHW under the BRAC UDP] told me that Rumel bhai will be able to do it for you as he also lives right next door. Then, we got the process done by him. He got us the form and didn't even take the money for printing the forms."

A 21-year-old woman from Slum A said that she learned about the vaccine registration process from a BRAC field officer. She explained:

> "There was a worker from BRAC in our community, one of the senior CHWs. I learned about the registration process from her. The Surokkha app, which has to be filled [in] online via Google; I have done vaccine registrations for many people using that app. I told them, 'If you want to do registration, please contact me. I will do registration for you.' People came to me, and I registered for them."

There were also youth-led volunteer clubs held in Slum A and Slum B, known as "Protimoncho" and "Youth Associates", who took the initiative to help people in their community with the vaccine registration process. These clubs had been actively raising awareness in their communities since the early days of the COVID-19 pandemic and were now focusing on COVID-19 vaccine awareness and registration support. A male from Slum A commented:

> "They [the Protimoncho] made the community people aware and told them to go there (*tara elekar manushjon ke oikhaney pathaise shocheton korse*). They instructed them to obtain the serial number [a code given to an individual, indicating their turn for vaccination] at 6 in the morning, then stand [in] the line this way during the mass vaccine campaign. This kind of support was provided by the youth groups."

### 3.3.2. Building Awareness and Vaccine Registration Support through CHWs and Community Groups

Respondents from all three sites mentioned that the BRAC CHWs have been supporting them with information on vaccines and updating them about the registration process regularly through the community yard meetings. A 45-year-old woman, living in Slum C with her daughter and grandchild, said:

> "The apas [CHWs] of BRAC, like Korima apa and Shaila apa [pseudonyms] discussed in the monthly meeting that those who want to take the vaccine, they should register. They will do the registration for us, and we will receive a message, seeing that we will have to go to the given address and take the vaccine. They need a voter ID/NID photocopy for this."

There were also other NGOs in Slums A and B working to raise COVID-19 vaccine awareness, as did the SAJIDA Foundation, but the BRAC was the only NGO working in Slum C. Another 28-year-old homemaker from Slum B commented:

"The SAJIDA people helped to do the entry then they said that you guys will have to go to Matrishodhon and take the vaccine."

However, in Slum C, there were Community Support Team (CST)[3] volunteers to help people with COVID-19 vaccine-related information and the registration process. A 20-year-old female from Slum C explained:

"In our area, CST is also [about] convincing people to register for vaccines. They visit households and support them to register."

### 3.3.3. Increased Willingness Regarding Vaccine Uptake, Influenced by Others in the Community

It was found that when someone in the community received the vaccine, others were also influenced by their action and were then vaccinated. A 35-year-old woman in Slum B, who runs a scrap metal shop with her husband, mentioned this practice of imitating others. She said:

"Many people were scared, as the vaccine is a new thing. Isn't it normal that people are afraid of trying anything new in the beginning? Once someone gets the vaccine, there is less fear. Some say that [the] disease will increase more, corona will never go. These are [wrong] beliefs shared by people. If one member from a family takes the vaccine, then other members of the family start to believe that [someone] who received the vaccine is in good health; therefore, if they take it, they will be able to stay healthy too."

Some of the younger respondents who were interviewed reported that they were motivated to receive the vaccine when they saw photos of other people receiving their vaccination or read their posts about getting vaccinated. One of the 23-year-old male NGO volunteers in Slum B explained:

"A lot of people on Facebook posted their photos after the COVID-19 vaccine and wrote that their first dose was given and encouraged others to get vaccinated. They also wrote that they did not face any problems after getting the vaccine dose, except for slight side effects like pain in the vaccine site and fever, which is similar to the side effects of most vaccines".

While social media posts and photos act as facilitators of vaccine uptake, they can also harbor fake news, which can increase people's hesitancy regarding vaccines.

### 3.3.4. The Push from Employers for Vaccination and the Importance of a Vaccine Certificate

Some respondents from Slum A and Slum B mentioned their receiving information and vaccine registration support from their employers. However, this varied, based on their different occupations. For instance, most females from Slums A and B who were working as housemaids had employers who helped them register for the vaccine. This is also because people living in urban slums are often stigmatized as carriers of COVID-19; thus, ensuring that they are vaccinated would mean that the members of the households in which they worked were better protected. People working in garment factories also received registration and vaccination support from their companies. However, as reported by a 20-year-old female from Slum A, most people from their community worked in smaller and less well-known garment factories near their area and had not received any support from their factories. She said:

"Apu, in our slum, most of the boys and girls from the younger generation are working in the garments [factories]. Garments factories which are renowned, e.g., Babylon, Apparel, they were supposed to give vaccines to the workers, but I

don't know whether they have already given it or not. But the smaller garments [factories] are not doing such things."

In Slum B, most people worked as waste collectors for the City Corporation. They were at increased risk of COVID-19 infections as they traveled widely to collect all kinds of waste, be it household waste, medical waste, etc. The City Corporation made vaccination mandatory for their staff and took steps to prioritize vaccination for their staff by providing necessary support. They also told their staff that they would not be receiving salaries if they were unable to show their vaccine certificates in the future. A 23-year-old male from Slum B, whose father worked as a waste collector for the City Corporation, shared how the local authority had supported the waste collectors by prioritizing their receiving the vaccine during the mass vaccination campaign.

He said, "In our area, it is mostly City Corporation workers. The commissioner, as well as the Conservancy Inspector at City Corporation, instructed that 10 people will be able to get vaccinated every day without standing in line. It was necessary for them to get the vaccine without standing in line because if they spent their time waiting for their serial number, who would do the work? Every day, in this way, 10 City Corporation cleaners were able to get vaccinated."

In Slum C, most people worked as day laborers or as factory workers in the nearby mills and factories. People working in informal jobs, for example, rickshaw pullers, construction site workers, and scrap metal collectors, received little or no support from their employers or work networks. There were few to no organizations, private enterprises, government bodies, or NGOs that had programs focusing on these more vulnerable occupation groups, which meant that they had limited information on vaccines and the registration process. In addition, taking a day off from work to receive their vaccinations would also mean losing one day's worth of their earnings, which was hard for them to afford.

There have been discussions in these communities that the vaccine certificate would be required in the future when applying for jobs or when seeking healthcare for major illnesses. A 45-year-old single mother from Slum C, who worked as a part-time home help and lived with her eldest daughter and grandchild, expressed her fear of not availing herself of treatment for underlying health issues because she feared that the hospitals would not be allowed to treat anyone without the vaccination cards. She said:

> "There has been this change because when the government first introduced this vaccine, we didn't know about it before, but now it has been discussed that there is a vaccine card. If you don't have that card, and you have some major health problems you cannot get admitted to the hospital. Then people cannot go to the hospital without the vaccine card. That's when I said, people with allergies/itchiness or diabetes, if they cannot get vaccinated, how will they receive the vaccine card?"

This perceived importance of the vaccine certificate and the associated economic consequence for the urban marginalized population was a major reason why so many people who were initially unwilling to receive the vaccine were eventually vaccinated. Despite these community mobilization efforts, the low uptake of the COVID-19 vaccine still exists, due to important systemic and other barriers.

*3.4. Systemic and Other Barriers to Vaccine Uptake*

3.4.1. Struggles with Online Vaccine Registration

Most respondents from all three study sites reported that they had faced challenges when filling out the online vaccine registration form as not all of them had access to smartphones. Those that did were not familiar with the process of going to an online website to register. A 35-year-old female from Slum B, working at a scrap (Bhangari) shop with her husband, mentioned failing to register her husband's vaccination. Another 40-year-old day laborer, a male respondent from Slum C, mentioned the same issue regarding not being able to understand the online registration process. He said:

"I have heard that people are taking [the] corona vaccine. There's a card through which people get the vaccine (*ekta card er madhomey corona tika tah nitase*). But we don't understand much about it."

Another 18-year-old female from Slum C who works for the community mentioned similar problems that were faced by marginalized people in her community. She said:

"At present times, a problem in our area is that vaccines are being provided through online registration and tokens. As a result, people in general are not getting vaccines".

### 3.4.2. Making Money by Charging the Poor for Online Vaccine Registration

Many found the registration process complex, and this further reduced their willingness to receive the vaccine. There have also been reports from all three sites of people trying to make money by charging poorer people a fee in return for helping them with the registration process. A 39-year-old female from Slum B mentioned people not being able to register online by themselves, as they are not used to such online processes and, hence, need support from local internet stores in return for money. She said:

"To speak of challenges, we were not able to fill out the form at home, so we went to the internet shop, and they helped us to fill [in] the form . . . the people from the internet computer shop in our neighborhood. They charged, like, 60 taka to help fill [in] the form."

For many, earning their livelihood that day outweighed the tedious vaccine registration process and the time needed to eventually take it. One 29-year-old female from Slum B, working as a part-time home help, said:

"If I spend a day [completing] this difficult registration form, I have to miss out on my work. I will also lose a day's work on the day I go to take the vaccine, and then if I get fever afterward, I will lose more days. Who will bear this loss for me?"

### 3.4.3. Irregularities during the Mass Vaccination Campaign by the Government

In August 2021, the Government of Bangladesh launched a mass vaccination campaign, wherein people could complete on-the-spot registration at any of the designated vaccination centers to receive the COVID-19 vaccine. This was particularly useful for urban marginalized people, who struggled with the online registration system. There was an allotment of 300 vaccines to be administered every day; many respondents reported that around two to three thousand people queued at the centers, which meant that many could not receive the vaccine during this campaign. People at all three sites lined up very early, just after their morning prayers, outside the ward councilor's office to collect tokens, then queued for the vaccine at a nearby mass vaccination center (usually a school near each designated ward). Most people reported that the number of vaccines allocated for each day was not enough for all the people who were living at the respective sites.

A 20-year-old female from Slum C, working as an NGO field staff member, said:

"Most people couldn't get the vaccine. People had to collect a token from the councilor's office. Every day, 250 tokens were being provided, which is not adequate."

Almost all respondents from Slum C mentioned that people who were affiliated with local and political leaders in their community were given priority during token distribution. Many mentioned that only a few people who had stood in line for tokens received them, while the rest were given to people with links to local leaders and those who were influential in the area. Another 40-year-old male day laborer from Slum C also mentioned that house owners in the area were also another group that was prioritized. He said:

"Those who are house owners are getting it (*elakar jara bariwala tara ashole beshi pay amra ashole pai nai akhono*) because the leaders got it first and as soon as they came

to the vaccine center, they got the vaccine (*maneh chairman office e awar shathey shathey tara agei pabo aetha ki abar kowan lagey naki*)."

In Slum C, a few people reported that during the mass vaccine campaign, when they went to collect their vaccination tokens, they needed to show their national ID cards; if they did not belong to that specific ward in which the slums are located, they were refused a token. The people at the ward councilor's office told them that only voters in that area were eligible for the vaccine. This meant that many who were registered as voters in their own villages, or in other wards of Dhaka city, were not able to access the vaccine. Mistrust in local administrative bodies is often found among marginalized communities. This stems from previous systemic stigmatization and discriminatory social systems and services; for example, irregularities during relief distribution during lockdown have caused many among the urban poor to believe that they would be left out of government vaccination campaigns.

Only a few people from Slum A and Slum B said that the local authorities, as with the ward councilors, were fair in their distribution of tokens and vaccines and prevented such irregularities from occurring during the vaccination campaign. At these sites, the elderly were also given priority and were brought to the front of the queue. In addition, many of the urban poor could not receive vaccines, despite their willingness to do so, when the mass vaccine campaign, which was supposed to run for seven days, was later shortened to three days due to vaccine shortages.

## 4. Discussion

Over time, more people from all three study sites were willing to be vaccinated. The findings in this study show that the low uptake of vaccines does not always indicate vaccine hesitancy; many participants reported wanting to be vaccinated but faced underlying barriers to receiving the vaccine. The context of Bangladesh and successful EPI programs are likely to underpin this willingness to be vaccinated. However, in this study, this was complicated by rumors and fears around the safety and efficacy of the vaccines that were being administered, along with misconceptions that the vaccine was only for children or was only for those who currently had COVID-19. A web-based study conducted among 605 urban and rural residents from Bangladesh reported similar findings: 61% of respondents wanted to be vaccinated; however, of these, only one-third were willing to receive it immediately (Mahmud et al. 2021). Our study found that rumors on social media platforms created panic and misconceptions among some readers, reducing people's willingness to be vaccinated, and that this challenging "infodemic" on social media platforms has been reported elsewhere (Demuyakor et al. 2021). Another study, conducted in urban Dhaka, found that it was the people who were not willing to receive the vaccine against COVID-19 who reported negative effects of the vaccine, such as dangerous side effects, death from vaccinations, etc. (Kalam et al. 2021). Past studies on influenza vaccines have shown that the safety of a newly developed vaccine is a major deciding factor in vaccination uptake (Schwarzinger et al. 2010).

Our study also showed that the perceptions and experiences of the COVID-19 vaccine varied across different occupations, with day laborers, such as rickshaw-pullers and street hawkers, being less likely to register for it or to be vaccinated, due to poverty, precarity, and lost wages. People involved in jobs that were paid monthly, such as garment trade workers and street cleaners, were more willing to be vaccinated as it was a mandatory requirement by their employers. However, the precarity faced by day laborers meant that they prioritized receiving their pay over spending hours in a queue waiting for their turn to be vaccinated. This issue is compounded by a lack of programs that focus on sensitizing and supporting these vulnerable occupation groups in accessing vaccines.

Our study identified several factors that influence the uptake of vaccines. Trust is key, and many people were more likely to accept vaccination if others in their community had been vaccinated. Many respondents in our study reported the significant role of their social networks—family, relatives, and neighbors—in motivating them to receive the vaccine. In addition, social media, television news programs, and occupation networks such as

colleagues and employers also played a role in encouraging residents to be vaccinated. The same result was found in a report by the BRAC Institute of Governance and Development (Faruk and Quddus 2021), which looked at data drawn from national-level surveys at different points in time. Health professionals and, particularly in the context of urban slums, CHWs had a great influence on people's acceptance of COVID-19 vaccines since they were a trusted source of information. Our study highlights the importance of bringing community views and trusted stakeholders into the process of co-creating and disseminating vaccine information and developing easily accessible systems and structures, to facilitate vaccine uptake.

Despite an increased willingness to be vaccinated, the majority of the respondents had not registered or were vaccinated. Women and the elderly, who had poor access to phones and internet connectivity, faced particular challenges. Studies have confirmed that fewer women than men receive COVID-19 information in Bangladesh as there is a 29% gender gap in mobile phone ownership and a 52% gap in internet use (Jahan and Zahan 2021). This is an area where community mobilization and awareness-raising activities by trusted community health workers and the young people in the community were useful, as this meant that more women living in urban areas received information and support related to the COVID-19 vaccines. In addition, many people received support and solidarity from people in their communities in terms of help with registration, which was often associated with a cost, such as paying a small sum of money or donating their time, in return for the assistance that they received.

The mass vaccination campaign in August 2021 was a way to address the barrier created by the online-based registration system. However, minor irregularities that led to the exclusion of people who were not registered as voters of Dhaka City were observed during the campaign, meaning that disadvantaged populations, such as the urban poor, were excluded (Transparency International Bangladesh 2021). It is important to account for these inequities when designing future vaccination campaigns.

Despite such irregularities, through the continuous efforts of government bodies, strong partnerships, and mobilization through NGOs, the media, youth volunteers, and systems that were put in place through employers, offices, and schools created a space that supported people in becoming vaccinated so that they could both continue with their economic activities and safeguard their families and others around them. Bangladesh and its COVID-19 vaccination drive have also been recognized as a success story by UNICEF in one of its recent publications (UNICEF 2022).

One limitation of this study was that all interviews were conducted by phone, in order to safeguard the research team and the participants since lockdown and other restrictions were imposed in Bangladesh at different times throughout the research period. This practice allowed us immediate access to certain participants while excluding those without phones. It also meant that non-verbal cues from the interviews could not be identified. However, we have been working closely in these communities under the umbrella of the ARISE project since 2019; through our discussions with NGOs and other stakeholders in the settlements, we were able to reach a wide range of vulnerable respondents. This approach also allowed us to assess the evolution of perceptions and attitudes during the various phases of vaccination among different slum dwellers in three different slum settings.

Our study details the trajectory of the COVID-19 vaccine scenario in urban slums, along with the different factors that play a role in ensuring vaccine uptake in these communities. We show that the willingness to be vaccinated varied through time, but it was also undermined by systemic barriers in the urban slum contexts of poverty and precarity. Raising vaccine awareness through the activities of community health workers, social media platforms, and different health services can be effective in addressing vaccine literacy. The collaborative and active involvement of social networks, the community, religious leaders and other influential people, and young people can contribute to motivating people toward vaccine acceptance. However, despite successful collaborative efforts from the government and NGOs to implement a successful vaccine rollout in this country, there

are still many challenges to achieving herd immunity. Understanding how the COVID-19 vaccine is perceived in urban slums and among diverse groups of the urban poor can help policymakers to design effective communication programs, along with systems and structures that enable adherence to the national vaccine rollout.

**Author Contributions:** Conceptualization, W.A., N.F. and F.M.; methodology, W.A.; validation, W.A. and F.M.; formal analysis, W.A., N.F. and F.M.; investigation, W.A., N.F. and F.M.; data curation, W.A., N.F. and F.M.; writing—original draft preparation, W.A., N.F. and F.M.; writing—review and editing, W.A., S.T. and S.F.R.; supervision, S.F.R.; funding acquisition, S.T. and S.F.R. All authors have read and agreed to the published version of the manuscript.

**Funding:** This research was funded by the UK Research and Innovation (UKRI). The GCRF Accountability for Informal Urban Equity Hub ("ARISE") is a UKRI Collective Fund award with award reference ES/S00811X/1. The funder had no role in study design, data collection, data analysis, data interpretation, or writing of the manuscript. For the purpose of open access, the author has applied a CC BY public copyright license (where permitted by UKRI, 'Open Government License' or 'CC BY-ND public copyright license may be stated instead) to any Author Accepted Manuscript version arising.

**Institutional Review Board Statement:** The study was reviewed and approved by the Institutional Review Board of the BRAC James P. Grant School of Public Health, BRAC University, Dhaka (IRB Reference No. 2019-034-IR).

**Informed Consent Statement:** Informed consent was obtained from all subjects involved in the study.

**Data Availability Statement:** Not applicable.

**Conflicts of Interest:** The authors declare no conflict of interest.

## Notes

[1]   In Bangladesh, the Government provides healthcare services to its rural people through health facilities called "upazila health complexes" (UHCs) at the upazila level and through union subcenters at the union level (the smallest administrative unit). This is the first referral health facility at the primary level of the healthcare delivery system in this country.

[2]   ARISE (Accountability for Informal Urban Equity Hub) is a GCRF UKRI-funded multi-country research project that began in February 2019. ARISE focuses on the health and wellbeing of residents of urban informal settlements in low and middle-income countries (LMICs). There is a total of eleven partner organizations from five countries, including Bangladesh, India, Kenya, and Sierra Leone, and it is led by the Liverpool School of Tropical Medicine (LSTM), United Kingdom.

[3]   The Community Support Team (CST) is an initiative by the Government of Bangladesh, several United Nations agencies (the FAO, UNFPA, UNICEF, the WFP, and the WHO), NGOs, and civil society organizations to control the spread of COVID-19 in communities across the country. Under this initiative, volunteer teams are deployed to urban slums to help identify those people with COVID-19 symptoms and support them with home-based case management, telemedicine support, hospital referrals, and diabetes and hypertension screening. It is funded by the World Bank Pandemic Emergency Funding Facility, USAID, and the Foreign, Commonwealth, and Development Office of the United Kingdom (FCDO).

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
