# Peer review of "Perceptions of and Attitudes toward COVID-19 Vaccination among Urban Slum Dwellers in Dhaka, Bangladesh"

_socsci, doi:10.3390/socsci12040230_

Round 1

Reviewer 1 Report

The choice of qualitative methodology is appropriate. However, the primary research data - interviews - is inadequately utilized. The author does not tease out the analytical content of the interviews fully. Rather there is a tendency for casual injection of unsourced hearsay - section 3.2.1 as example - and sweeping statements such as ' These created mass fear among the communities which further led them to be hesitant to register for the vaccines'. The strength of the article lies in the evolution of perceptions and attitudes during various phases of vaccination. While these are described well, the author's analytical lens is unnecessarily and overly tied to the vaccine hesitancy thesis. 

Author Response

Response to Reviewer 1 Comments

Point 1: The choice of qualitative methodology is appropriate. However, the primary research data - interviews - is inadequately utilized. The author does not tease out the analytical content of the interviews fully. Rather there is a tendency for casual injection of unsourced hearsay - section 3.2.1 as example - and sweeping statements such as 'These created mass fear among the communities which further led them to be hesitant to register for the vaccines'. The strength of the article lies in the evolution of perceptions and attitudes during various phases of vaccination. While these are described well, the author's analytical lens is unnecessarily and overly tied to the vaccine hesitancy thesis. 

Response 1: Thank you for this thoughtful comment. We agree that the strength of the paper lies in evolving attitudes and perceptions through the different phases of the vaccination process, and we have made this clearer both in the methods and the discussion. We have also brought additional nuance out from the interviews to better illustrate the depth and detail of responses that emerged through the analytical process and to avoid sweeping statements and bring more of a critical lens to the vaccine hesitancy thesis. For example: In section 3.2.1, we have edited the statement 'These created mass fear among the communities which further led them to be hesitant to register for the vaccines' to ‘These rumours created fear amongst some slum community members, bringing anxiety and leading them to be hesitant to register for the vaccine’ (Pg 9). We have also made minor edits to Table 2 (Pg 5) describing the respondent categories so as to avoid stereotyping. We have tried to be more critical about the use and application of vaccine hesitancy by we have pulled out the experiences and challenges faced by many urban slum dwellers with regards to COVID-19 vaccination, that have also led to low uptake. We have tried to define ‘vaccine hesitancy’ in the context of vaccine uptake and the different factors that influence willingness. We have emphasized throughout the paper and reiterated in the conclusion how willingness varied through time and was influenced by factors such as poverty, misinformation and fear.

Reviewer 2 Report

·      Overall, this is a neat paper with adequate literature review, a qualitatively rich study based on a small sample, well substantiated findings and discussion. The comments that I have are relatively minor and aimed at further enriching the paper.

·      Literature review: It will be useful to give some idea of the overall state of urban services and particularly, the primary health care system that exists in the slums of Bangladesh. It is also useful to give some introduction to the way in which COVID 19 vaccines were rolled out in cities of Bangladesh and the nature of access to slum residents.

·      The choice of sample is multi-stage and provides for heterogeneity and a consideration of the most vulnerable groups.

·      The backdrop to the selection of slums gives an idea of NGO services that existed in the three slums but fails to give any indication of ‘state services’. Also does existence of services mean that all households in the particular settlement have access to those services? The sampling criteria mentions that the three settlements were chosen for their heterogeneity but no significant differentiators except the geographical location and population are mentioned. These differentiators may need to be specifically mentioned. Further, in the findings; there is no discussion that mentions any significance of the heterogeneity of the settlements. Does this mean that location did not make a significant difference?

·      A very interesting finding of the paper is that it highlights the importance of systemic and other barriers in large scale vaccine uptake as opposed to hesitancy. It also illustrates what strategies at various levels have been useful in enabling vaccine uptake. At one level, these findings stand in stark contrast to one of the studies quoted in the literature review from Bangladesh which suggests that people in slums do not want to take vaccines. At another, perhaps the authors could critique the concept of hesitancy itself and propose a new concept. After all, the purpose of qualitative research is to propose new constructs that may transform dominant discourses.
